# Cardiac prehabilitation, rehabilitation and education in first-time atrial fibrillation (AF) ablation (CREED AF): Study protocol for a randomised controlled trial

Nakul Chandan[1,2], Violet Matthews[1,3], Hejie He[1,4], Thomas Lachlan[1,4], Ven Gee Lim[1,4], Shivam Joshi[1,3], Siew Wan Hee[1,3], Angela Noufaily[4], Edward Parkes[1,4], Shilpa Patel[1,4], Lazaros Andronis[4], Joanna Shakespeare[1,4], Helen Eftekhari[1,4], Asad Ali[1], Gordon McGregor[1,2,4], Faizel Osman[1,2,4]*

1 Institute for Cardio-Metabolic Medicine, University Hospital Coventry, Coventry, United Kingdom, 2 Centre for Healthcare & Communities, Coventry University, Coventry, United Kingdom, 3 Research and Development Institutes of Excellence, University Hospital Coventry, Coventry, United Kingdom, 4 Warwick Medical School, University of Warwick, Coventry, United Kingdom

* faizel.osman@uhcw.nhs.uk

**Data Availability Statement:** No datasets were generated or analysed during the current study. All relevant data from this study will be made available upon study completion.

## Abstract

### Background

Atrial fibrillation (AF) is associated with significant morbidity/mortality. AF-ablation is an increasingly used treatment. Currently, first-time AF-ablation success is 40–80% at 1-year, depending on individual factors. There is growing evidence for improved outcomes through management of AF risk-factors/comorbidities via patient education/exercise-rehabilitation. There are no studies assessing combined prehabilitation/rehabilitation in this cohort. The aim of this randomised controlled trial is to assess efficacy of comprehensive prehabilitation/ rehabilitation and combining supervised exercise-training with AF risk-factor modification/ education compared with standard care in people undergoing first-time AF ablation.

### Methods

This single-centre pragmatic randomised controlled trial will recruit 106 adults with paroxysmal/persistent AF listed for first-time AF-ablation. Participants will be randomised 1:1 to cardiac prehabilitation/rehabilitation/education (CREED AF) intervention or standard care. Both groups will undergo AF-ablation at 8-weeks post-randomisation as per usual care. The CREED AF intervention will involve 6-weeks of prehabilitation (before AF-ablation) followed by 6-weeks rehabilitation (after AF-ablation) consisting of risk factor education/modification and supervised exercise training. Standard care will include a single 30-minute session of risk-factor education. Outcomes will be measured at baseline, 10-weeks and 12-months post AF-ablation, by researchers blinded to treatment allocation. The primary outcome is cardiorespiratory-fitness (peak oxygen uptake, $VO_{2peak}$) assessed using cardiopulmonary exercise testing (CPET) at 10-weeks post-ablation. Secondary outcomes include health-related quality of life, AF recurrence/burden assessed by 7-day Holter-monitor, requirement

**Funding:** Boston Scientific are pleased to confirm the funding of the research grant for CREED AF. As outlined in the agreement, Boston Scientific Limited shall pay the Grant Recipient, the total grant sum of £297,191.37 provided that funding reports are satisfactory and received as outlined.

**Competing interests:** The authors have declared that no competing interests exist.

**Abbreviations:** AF, atrial fibrillation; AFEQT, AF Effect on Quality-of-Life Questionnaire; CEAC, cost-effectiveness acceptability curve; CI, confidence interval; CONSORT, Consolidated Standards of Reporting Trials; CPET, cardiopulmonary exercise testing; CREED AF, cardiac prehabilitation, rehabilitation and education in first-time AF ablation; DCCV, direct current cardioversion; GCP, good clinical practice; HRQoL, health-related quality of life; ICER, incremental cost-effectiveness ratios; ICHOM, International Consortium for Health Outcomes Measurement; MACE, major adverse cardiovascular events; NHS, UK National Health Service; PIS, patient information sheet; PVI, pulmonary vein isolation; QALY, Quality-adjusted life year; SD, standard deviation; SF-36, 36-Item Short Form Survey.

for repeat AF-ablation, study defined major adverse cardiovascular events, and cost-effectiveness (incremental cost per quality-adjusted life year (QALY)).

## Conclusions

This study will assess clinical-efficacy/cost-effectiveness of comprehensive prehabilitation/rehabilitation/patient-education for people undergoing first time AF-ablation. Results will inform clinical care and design of future multi-centre clinical trials.

## Trial registration

**URL:** https://www.clinicaltrials.gov; Unique identifier: NCT06042231.Date registered: September 18, 2023.

## Background and rationale

Atrial fibrillation (AF) is a common arrhythmia [1–3] and is independently associated with increased morbidity and all-cause mortality [3–8]. Two-thirds of people with AF have at least intermittent symptoms which can be disabling and can markedly impair health-related quality of life (HRQoL) [9, 10]. Furthermore, AF is a growing epidemic due to the ageing population, chronic cardiovascular diseases, and the accumulation of AF related risk factors, such as diabetes, obesity, hypertension, alcohol, and smoking [6, 11, 12]. Current AF management is largely focused on stroke prevention via anticoagulation, and heart failure prevention with ventricular rate or rhythm control strategies (anti-arrhythmic drugs and catheter or surgical ablation) [3, 13] and is a top priority in the UK National Health Service (NHS) long-term plan [14]. Although current strategies can be effective in the short term, long-term success is limited and there is a failure to address key patient outcomes including exercise capacity and HRQoL. Invasive catheter ablation is an increasingly common treatment option for rhythm control in AF. The success rate is ~40–80% at 12 months, depending on the AF subtype and other clinical factors, but there is considerable attrition over time with AF recurrence at longer term follow-up [3, 13].

Cardiac rehabilitation is an integral part of long-term cardiovascular disease management, reducing cardiovascular mortality and hospital re-admissions, and improving HRQoL [15, 16]. Exercise training, education and risk factor management are core components [17–21]. At present, people with AF do not routinely receive cardiac rehabilitation as part of standard care. Nurse-led patient education and risk factor modification programmes have been shown to reduce AF burden and hospitalisations, whilst improving medication adherence and HRQoL [22]. Prospective studies have demonstrated that active management of AF related risk factors can lead to improvement in arrhythmia-free survival after catheter ablation [3, 23]. In recent years, studies investigating the impact of multi-disciplinary cardiac rehabilitation in people who have undergone catheter AF ablation have reported positive physical and mental health outcomes [24].

Prehabilitation (exercise training and patient education *before* a scheduled intervention) has been shown to benefit people who are undergoing surgery/cancer treatments, with fewer post-operative complications, less post-operative pain, and reduced length of hospital stay [25–27]. A recent systematic review exploring prehabilitation prior to major non-cardiac surgery reported prehabilitation reduced overall morbidity following surgery [25]. In cardiac

populations, other than people with AF, prehabilitation demonstrated beneficial effects post-procedure [25–28], however, there are no studies investigating prehabilitation in people undergoing AF ablation. It is a not known if a comprehensive programme of targeted prehabilitation, rehabilitation consisting of supervised exercise training and AF risk factor education/modification, can improve clinical and health-related outcomes for people undergoing first-time AF catheter ablation, or if there are any long-term benefits or harms. The aim of this this single centre randomised controlled trial is to test the clinical efficacy and cost-effectiveness of the Cardiac prehabilitation, rehabilitation and education (CREED AF) intervention, compared with standard care for people undergoing first-time AF catheter ablation.

## Methods

### Study design

This protocol follows guidance from the Consolidated Standards of Reporting Trials (CONSORT) [29]. A CONSORT checklist is provided as an Additional File. **Fig 1** shows the SPIRIT schedule. CREED AF is a single-centre, pragmatic randomised controlled trial comparing the impact of a cardiac prehabilitation and rehabilitation programme including supervised exercise training and risk factor education/modification, (i.e. alcohol intake, smoking, hypertension, diabetes, exercise, sleep apnoea), with usual care in people undergoing first time catheter AF-ablation (**Fig 2**). The randomisation allocation ratio will be 1:1.

### Ethics approval and consent to participate

The trial design and research protocol have been approved by the West Midlands-Solihull Ethics Committee, UK (23/WM/0125). The trial will be conducted in accordance with Good Clinical Practice, UK Law, and the Declaration of Helsinki 2002. Written, informed consent to participate will be obtained from all participants.

### Study population, participant identification, recruitment, and informed consent

People who have been listed for first-time AF ablation and meet the eligibility criteria (**Fig 3**), will be identified from specialist arrhythmia clinics at University Hospitals Coventry and Warwickshire NHS Trust, Coventry, UK and approached either face-to-face during a clinic appointment, or remotely via telephone and/or by post. Potential participants will be provided a participant information sheet (PIS) and invitation letter. Those expressing interest in participating will be invited to attend a baseline assessment, with eligibility reconfirmed and informed consent obtained in writing.

### AF ablation pathway

Participants are only eligible for the CREED AF trial if they have been listed for AF ablation. AF ablation is not part of the CREED AF intervention. Normal local clinical practice and procedures will be followed. Briefly, people will be identified for catheter AF-ablation by a cardiologist specialising in arrhythmia management. Depending on the AF sub-type, any cardiac imaging findings (i.e., ventricular and/or atrial cardiomyopathy) and burden of co-morbidities, people will be informed that success of first AF ablation procedure is 40–80% (70–80% for paroxysmal AF, 40–60% for persistent AF) at 12 months. Therapeutic oral anticoagulation will be initiated (if not already in place) and taken reliably for at least 21-days prior to the procedure. Structured risk factor management is not routinely implemented in clinical assessments. A review of medications, along with bloods tests, and 12-lead ECG will take place during pre-

| Procedure | Pre-Screening/ Lead-In Period | Consent & Baseline | Pre-Ablation Intervention | AF Ablation Procedure* | Rest | Post-Ablation Intervention | Follow-Up & Outcome Measure | Follow-Up |
|---|---|---|---|---|---|---|---|---|
| | Month -4 to Month -1 | Week -2 to Day 1 | Up to Week 8 | Week 9 (Day 1 Ablation) | Up to Week 11 (2 weeks post-ablation) | Up to Week 19 (Up to week 10 post-ablation) | Up to Week 21 (Up to week 12 post-ablation) | Month 12 (±4weeks) post ablation |
| Screening | X | | | | | | | |
| Eligibility assessment | X | | | | | | | |
| Re-Confirm Eligibility | | X | | | | | | |
| Informed consent | | X | | | | | | |
| Questionnaire completion (EQ-5D, SF-36, AFEQT) | | X | | X | | | X | X |
| Resource Use Questionnaire | | | | X | | | X | X |
| Demographic data (Year of birth, sex, ethnicity, height, and weight) | | X | | | | | | |
| Smoking and diet history | | X | | | | | | |
| Relevant clinical history | | X | | | | | | |
| Current medications | | X | | | | | | |
| Standard blood tests* | | X | | | | | | |
| ECG* | | X | | | | | | |
| Cardiopulmonary exercise testing | | X | | | | | X | |
| Randomisation | | X | | | | | | |
| 1-to-1 behavioural & motivational education sessions | | | X[1] | | | X[1] | | |
| 30-minute AF risk factor modification education session | | | X[2] | | | | | |
| AF Ablation* | | | | X | | | | |
| Cardiopulmonary exercise sessions for intervention arm | | | X | | | X | | |
| Adverse Event Reporting | | X | X | X | X | X | X | X |
| 1–7-day(s) Holter monitor | | | | | | | X | X |
| Primary Outcome Analysis | | | | | | | X | |
| Review of hospital records (MACE, need for redo-AF ablation, etc.) | | | | | | | | X |

* Part of Standard Care

X[1] 1-to-1 behavioural & motivational education session alternating weeks for 6-8 weeks pre- and post-ablation only occurring in the intervention arm

X[2] one off 30-minute education session activity only occurring in the control arm

**Fig 1. Trial schedule of events (SPIRIT schedule).**

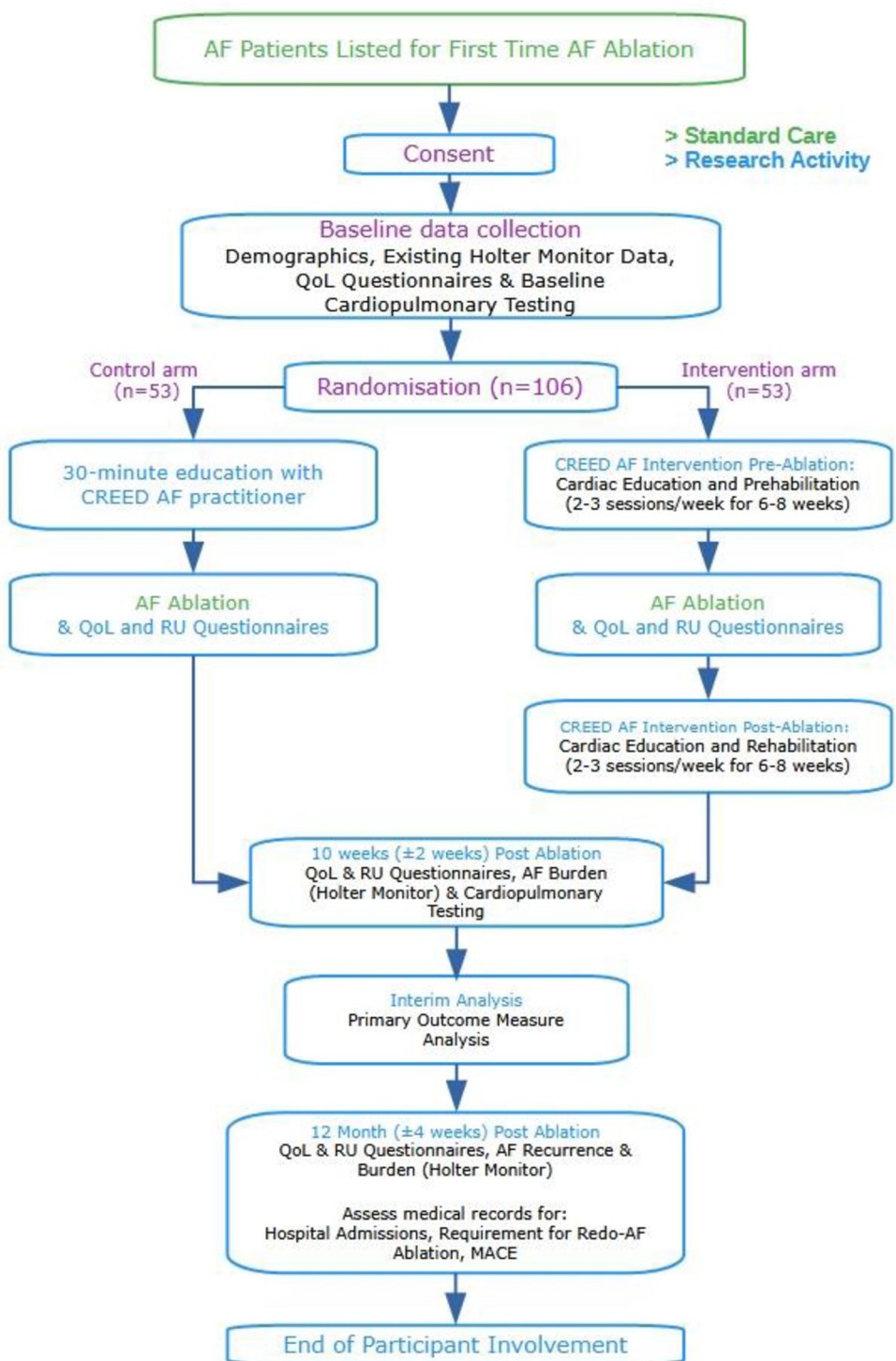

**Fig 2. Flowchart of the CREED AF study, demonstrating the patient pathway.** AF, atrial fibrillation; QoL, quality of life; RU, resource use; MACE, major adverse cardiovascular events.

Inclusion Criteria:

 a) all patients listed for first-time AF ablation

 b) age ≥18 years

 c) can access online exercise and support sessions from home

Exclusion Criteria:

 a) pregnancy

 b) lack of capacity to consent and participate

 c) presence of contraindications or limiting physical or mental health co-morbidity preventing travel, safe exercise, or productive engagement with the trial procedures

**Fig 3. Eligibility criteria.**

procedure assessment (see **Fig 1**). People may undergo electrical direct current cardioversion (DCCV), as a temporising procedure, especially if in persistent AF and highly symptomatic. People undergoing first time AF-ablation have pulmonary vein isolation (PVI) only as the initial strategy for both paroxysmal and persistent AF. This will be done using either cryoablation or radio-frequency ablation technology; pulse-field ablation is currently not available at our centre but could be included if it becomes available. Routine outpatient clinical review takes place at 6-months post-ablation. If there is a persistent episode of AF in the first 3 months post-ablation (during the 'blanking period') people will be listed for DCCV to restore sinus rhythm. If there are further episodes of AF after the blanking period, the AF treatment strategy will be re-considered, and may include listing for a redo-AF ablation, performed using 3D mapping/ablation. Further follow-up will depend on the treatment strategy agreed between the clinician and the patient. All participants will undergo first-time AF ablation with isolation of the pulmonary veins only as described above. Data will be collected on details of their AF ablation procedure, including energy type and duration of lesions. Any procedural complications will be recorded.

## Randomisation and blinding

After completion of baseline assessments, randomisation will occur using the integrated randomisation generator from Castor Electronic Data Capture [30]. The randomisation sequence is based on variable pre-specified block randomisation. Participants will be stratified by sex (male/female), AF sub-type (paroxysmal/persistent), left ventricular ejection fraction (<50%/≥50%), and amiodarone use (yes/no). Participants and CREED AF practitioners will be notified of the treatment allocation. Outcomes assessors will be blind to group allocation. We will request that participants do not reveal their treatment allocation and will record if they do so.

## Control group: Standard care

Participants will be invited for a single 30-minute one-to-one education session with a CREED AF practitioner (specialist nurse or doctor) prior to their AF ablation. This will be focused only on exploring the impact of AF related risk factors guided by freely available information leaflets from reliable charity organisations (Arrhythmia Alliance, https://heartrhythmalliance. org/aa/uk/resources/arrhythmia-alliance-patient-resources).

## Intervention group: CREED AF intervention

The CREED AF intervention will involve a comprehensive programme of structured exercise training and risk factor modification delivered both before AF-ablation (prehabilitation) and after AF-ablation (rehabilitation). To ensure generalisability to the UK NHS, the underpinning framework of the CREED AF exercise intervention is based on UK cardiopulmonary rehabilitation guidelines and service delivery models [31]. To maximise accessibility, exercise will be delivered as a 'rolling' programme. Participants can immediately join existing cardio-pulmonary rehabilitation exercise programmes rather than waiting for the recruitment of sufficient numbers to form a discrete group. The CREED AF intervention has 4 components:

**Component 1: Individual assessment and exercise familiarisation.** Participants will be invited to attend an initial one-to-one appointment with a CREED AF practitioner for clinical assessment and appraisal of participant goals, expectations, fears, concerns, and barriers to change for modifiable risk factors. During the first week of the intervention, a one-to-one supervised gym-based familiarisation exercise session will enable participants to build their confidence and allow practitioners to optimise exercise prescription.

**Component 2: Supervised exercise programme.** Familiarisation will be followed by a tailored, individualised, exercise training programme comprising 2–3 times weekly exercise for 6–8 weeks before, and 8–12 weeks after AF ablation (see **Fig 2**). A two-week intervention free period immediately post AF-ablation will allow for femoral access site wound healing. Exercise will be a combination of conventional gym exercise and functional fitness training and will be highly adaptable to allow for lower or higher ability participants, whilst ensuring safety and efficacy. Participants will have access to both in-person sessions at the local cardiac rehabilitation centre and instructor-led, live home exercise sessions that will take place in groups via video conferencing software.

**Component 3: AF risk factor education and modification.** A comprehensive programme centred around the education, and modification of AF risk factors will be delivered in one-to-one sessions with a qualified member of the research team either face-to-face or via video conferencing. The content will be delivered every second week (3 prehabilitation sessions and 3 rehabilitation sessions) before or after exercise as part of one-to-one 30-minute sessions. Specific AF risk factors will be targeted and will include support with smoking cessation (nicotine replacement therapy), reducing/stopping alcohol intake, advising on optimal blood pressure and diabetes control, and the clinical assessment for presence of sleep apnoea (using the Epworth Sleepiness Scale, and referring for consultation down the appropriate diagnostic/management pathway), in addition to advice optimising weight and performing regular moderate exercise as part of the rehabilitation intervention. Further written material, including a participant workbook and publicly available AF information leaflets will be provided together with signposting to relevant support groups.

**Component 4: Motivational support.** Recognising the importance of psychological support in people with AF, the comprehensive AF risk factor modification sessions will be supplemented by discussions which incorporate motivation to change, barriers to change, goal setting and problem-solving to build self-efficacy and encourage concordance with risk factor

modification. The behavioural and motivational sessions will aim to help improve short- and long-term adherence to exercise and risk factor modification goals.

## Outcomes

In line with the International Consortium for Health Outcomes Measurement (ICHOM) standard of outcome measures for AF [32], the primary outcome is aerobic capacity as determined by relative peak oxygen consumption ($VO_{2peak}$) measured by CPET at 10±2 weeks post-ablation (**Fig 2**). This time interval was chosen to ensure the 3-month blanking period was excluded in the reassessment. The CPET will be performed according to national professional body standards in a UKAS accredited respiratory lab [33]. The local standard operating procedure for CPET will be followed.

Secondary outcomes, measured at 10 weeks and 12 months post-ablation, include HRQoL as determined by the AF Effect on Quality-of-Life Questionnaire (AFEQT) [34] and 36-Item Short Form Survey (SF-36) [35]. Recurrence of AF post-ablation, as well as assessment for AF burden and requirement for redo AF ablation, will be evaluated with 7-day cardiac Holter monitors, and on review of participant medical records at 12 months post-ablation. All AF recurrence (defined as AF lasting ≥30-seconds) [3], AF/AT burden, and time of occurrence will be recorded. At 12 months post-ablation, medical records will be used to review health service activity for major adverse cardiovascular events (MACE): hospital admission related to AF, heart failure hospitalisation, new heart failure diagnosis, non-fatal myocardial infarction, non-fatal stroke, cardiovascular death, and all-cause mortality.

As part of the embedded economic evaluation, quality adjusted life years (QALYs) will be calculating using participants' responses to the EQ-5D-5L instrument [36, 37], which comprises a visual analogue scale (EQ VAS) and a health status descriptive system. The latter asks respondents to indicate their health state by ticking boxes next to the statement that represents the level of health (no problems, slight problems, moderate problems, severe problems, and extreme problems) across five dimensions (mobility, self-care, usual activities, pain/discomfort, and anxiety/depression). Each participant's responses to the EQ-5D-5L will be translated into a single, preference-based HRQoL index score (utility value) using recommended UK value sets at the time of the analysis. QALYs will be calculated as the area under the curve connecting utility values reported at different time points in time [38].

Resource use and relevant costs will include the cost of implementing and delivering the intervention, further use of NHS care, patients' out-of-pocket expenses and time-related costs captured using resource use questionnaires on the day of ablation (pre-ablation period), at 10-week post-ablation, and 12-month post-ablation. Local medical records, primary care records and data from other hospitals will also allow for accurate recording of health service activity. Adverse events will be recorded as per good clinical practice (GCP) guidelines.

## Study design feedback

Prior to submission to the research ethics committee, the study protocol and other patient facing documents have been reviewed by expert members in the field of cardiology and ablation. The study was also presented to the Patient and Public Research Advisory Group (PPRAG) at UHCW to receive expert lay feedback on the study design and methodology. Following engagement with the PPRAG at UHCW, they were supportive of this project. They recommended that people who do not have access to video conferencing software (smartphones or webcams) be excluded due to the reliance on the live exercise sessions at home in the CREED AF intervention. The group were also happy to see the familiarisation period incorporated into the programme.

## Sample size

The sample size calculation was based on the analysis of the change in the primary outcome (relative $VO_{2peak}$) at 10-week post-ablation from baseline. Previous studies investigated $VO_{2peak}$ at 6 months post-ablation, which we assume to be similar at 10-week post-ablation. Based on Kato et al, we can assume that the $VO_{2peak}$ standard deviation is around 3ml/kg/min at baseline and 4ml/kg/min at 10-week post-ablation [23]. As the correlation between the time points is unknown, we assume a value of r = 0.5. Consequently, the standard deviation of the difference from 10-week post-ablation to baseline is around 3.6ml/kg/min. Based on Fiala et al and Mujovic et al, we can assume that the change in $VO_{2peak}$ will be around 2ml/kg/min due to AF ablation in both AF-types, and a further 4ml/kg/min due to cardiac rehabilitation (which comprises pre- and post-AF ablation rehabilitation) [23, 39, 40]. Therefore, the difference between the intervention arm and standard care is expected to be 4ml/kg/min at 10-week post-ablation compared with baseline. To detect a conservative difference of 2.5ml/kg/min (an increase in 2.5ml/kg/min $VO_{2peak}$ in the intervention arm from standard care at 10-week post-ablation compared with baseline) at 5% significance level, and 90% power, we require 88 participants. To allow a drop-out rate of 20%, we will recruit 106 people, with 53 randomly allocated to each group.

## Data analysis

Demographics and baseline characteristics categorical data will be summarised as frequency and percentage, continuous data will be summarised using the following descriptive statistics: frequency (total number of missing and non-missing available for summary), mean and standard deviation (SD), median, 25th and 75th percentiles, minimum and maximum by treatment arm and as overall.

The primary outcome measure will be the difference in means between the two treatment arms by fitting a linear regression model where the relative $VO_{2peak}$ at 10-weeks post-ablation is the response, and the predictors are the relative $VO_{2peak}$ at baseline and treatment arm. We will adjust for other baseline measures, e.g., AF type, sex, left ventricular ejection fraction and amiodarone use. All participants with both relative $VO_{2peak}$ at baseline and 10-week post-ablation are the target population with those assigned to standard care through randomisation will be the control group and those to CREED AF intervention will be the active intervention group. The intercurrent event that we expect is treatment discontinuation due to any reason and this will be addressed as treatment policy strategy. We may also explore for any potential confounders in relative $VO_{2peak}$ within the study, and the complier averaged causal effect (CACE) analysis as a sensitivity analysis.

For secondary outcomes: all the scoring of HRQoL (AFEQT and SF-36) will be computed as per the instrument algorithm. The scores will then be summarised descriptively overall and by treatment arms at baseline and the short- (10-week) and long-term (12-month) follow-up time points. The change of HRQoL from baseline to 10-week, from baseline to 12-month post-ablation, and the difference between treatment arms will be estimated with their associated 95% confidence intervals (CIs). We will explore the HRQoL longitudinal effect by visualising the change against time. Proportion of significant AF events by treatment arms will be estimated with associated 95% confidence interval (CI) using Kaplan-Meier analysis. Similarly, the proportion of individual and composite MACE and its associated 95% CI by treatment arms will be estimated. The recurrence of AF burden from the Holter and each of the MACE occurrence will also be estimated by treatment arms as a time-to-event estimand. We will also perform a subgroup analysis by types of AF (persistent and paroxysmal AF) for all endpoints. Analysis will be performed with the appropriate statistical software and package available at

time of analysis. The statistical analysis plan will be finalised prior to primary outcome analysis.

## Health economic analysis

The economic analysis will be conducted on an 'intention to treat' basis. Missing data will be accounted for by using appropriate techniques, such as multiple imputation, depending on the extent and type of missing items. As cost distribution is skewed by existence of people with very high costs, calculated mean per-person costs will be provided with confidence intervals obtained through non-parametric bootstrap methods [41]. We will perform an incremental analysis to calculate difference in costs and in outcomes associated with the compared options. The findings will be presented as incremental cost-effectiveness ratios reflecting the extra cost for additional units of outcome. A sensitivity analyses will be done to assess impacts of different assumptions on results [42]. Due to inherent uncertainty secondary to sampling variation, the joint distribution of differences in cost and outcomes (QALYs) will be derived by performing non-parametric bootstrap simulations [43]. The simulated costs and outcome pairs will be depicted on a cost-effectiveness plane and plotted as cost-effectiveness acceptability curves (CEAC). CEAC will show the probability of CREED AF being cost-effective across a range of values of willingness to pay for an additional QALY [44].

## Adverse event management

In our study an adverse event (AE) is defined as any untoward medical occurrence involving a participant, which does not necessarily have a causal relationship with the intervention or trial. Adverse events will be collected only once the participant has provided consent and enrolled into the study. Expected AEs, related to the exercise outcome assessments or the exercise intervention, include 'normal' levels (for each individual) of: breathlessness, light-headedness/dizziness, muscle and joint stiffness/soreness, and tiredness/fatigue. There are a few AEs which are expected following AF ablation and occur after roughly 2–4% of all AF ablation cases, including: vascular injury, pericardial effusion/tamponade, stroke/transient ischaemic attack, myocardial infarction, phrenic nerve injury, atrioventricular (heart) block, and pulmonary vein stenosis. Because of the post-AF ablation conditions, there is a small risk (1:400) of requiring further cardiac surgery or a blood transfusion. Each AE will be assessed by the Chief Investigator, or a delegated clinician, to assess the causality to the CREED AF intervention. In the unlikely event that it is decided that one of the above post-AF ablation conditions is related to the CREED AF intervention, this will be reported as such in line with local Trust policy and national safety reporting guidelines. Expected AEs will be recorded on the participants' case report form (CRF) (AE form). Any unexpected AEs related to the exercise outcome assessments or the exercise intervention for all participants will be recorded on the CRF (AE form), logged, and reported at trial management group (TMG) meetings. Should multiples of the same unexpected AE occur during the trial period, then this will trigger the TMG to review the study procedures and consider any changes that need implementing. For the purposes of the study, serious adverse events (SAEs) will be an untoward medical occurrence that fulfils one or more of the following: results in death, is life-threatening, requires hospitalisation or prolongation of an existing hospitalisation, results in persistent or significant disability or incapacity, requires medical intervention to prevent one of the preceding, or is otherwise considered medically significant by the investigator. Though all admissions to hospital are an SAE, the following are expected within the study and do not require additional reporting but must be recorded in the CRF: disease progression (worsening symptoms/AF attrition unrelated to the exercise intervention), treatment (elective or pre-planned for a pre-existing condition) not

associated with any deterioration in condition, and general care not associated with any deterioration in condition. SAEs will be collected from self-report and from medical records checks. Reportable SAEs in both groups will be any event occurring within 24 hours of each assessment appointment (baseline and 10 weeks (±2 weeks) post-AF ablation), and in the intervention group any event that occurs at any time between baseline outcomes assessment and the 10 weeks (±2 weeks) post-AF ablation. All SAEs will be recorded and reviewed by the Chief Investigator to determine if it is directly attributable to the intervention and investigated in line with the hospital Trusts' standard operating procedure (SOP). Those SAEs that are deemed to be unexpected, probably, or definitely related to the trial interventions or outcomes assessments, will be notified to the Research Ethics Committee within 15 days, and reported to the TMG at the next meeting.

## Discussion

AF is an increasingly prevalent arrhythmia independently associated with morbidity and mortality.[3] In addition to pharmacological management, catheter AF-ablation is a recognised treatment option. However, success rates for AF ablations are suboptimal and AF often recurs at longer-term follow up. This is often due to co-morbidities and AF risk factors. The benefits of supervised exercise training and AF risk factor management are increasingly being recognised. Performing a relatively inexpensive prehabilitation and rehabilitation intervention (CREED AF) consisting of supervised exercise training and AF risk factor education/modification may improve outcomes for people undergoing AF ablation. Prehabilitation and rehabilitation are not routinely recommended for AF and emerging evidence has not been evaluated in large clinical trials. Exercise training and risk factor modification are of benefit in other cardiac populations and therefore could be of benefit to people with AF. The CREED AF study will indicate whether larger multi-centre trials for people with AF are warranted, informing a potentially clinically and cost-effective programme in line with NHS priorities for the long-term management of AF. These findings will also be of interest to healthcare providers globally.

The question about athletes, particularly high-endurance level athletes, was considered when planning and developing the research protocol for CREED AF. The general consensus was to maintain generalizability of the study by being inclusive of as many individuals suffering with AF undergoing first time AF ablation. It has been observed that high-endurance athletes do also suffer with AF, with reduced physical ability/fitness prior to ablation, and improvements in levels of fitness/exertion after AF ablation [45]. The application of CREED AF intervention, which aims at delivering a moderate-level of exercise (tailored to the individual), alongside risk factor management, is equally applicable in these participants. Furthermore, the primary outcome measure is mean change in $VO_{2peak}$ from baseline to primary follow-up at 10+/-2 weeks post-AF ablation in the two groups; measuring and comparing the change within the same individuals will minimize any bias, if any, derived by inclusion of athletes.

### Trial status

The CREED AF study started recruitment as of December 2023 and is expected to finish recruitment by February 2025.

### Monitoring, audit & inspection

The study may be monitored by the Research & Development Department at UHCW as representatives of the Sponsor, to ensure that the study is being conducted as per protocol, adhering to Research Governance and GCP.

## Committees

The trial steering committee (TSC) will consist of exercise, respiratory and arrhythmia specialists, and a member from the Patients and Public Engagement Committee (who have informed this study protocol). This group will meet regularly to review the progress of the study in terms of recruitment, endpoints, and adverse events. The publication of results will be overseen by the TSC. There is no data monitoring committee (DMC).

## Supporting information

**S1 Checklist. SPIRIT 2013 checklist: Recommended items to address in a clinical trial protocol and related documents\*.**
(DOC)

**S1 File.**
(DOCX)

## Acknowledgments

We thank the Research and Development Department at UHCW NHS Trust, and the team in Cardiac Rehabilitation for their support.

## Author Contributions

**Conceptualization:** Gordon McGregor, Faizel Osman.

**Data curation:** Nakul Chandan, Violet Matthews, Siew Wan Hee, Faizel Osman.

**Formal analysis:** Nakul Chandan, Siew Wan Hee, Angela Noufaily, Lazaros Andronis, Faizel Osman.

**Funding acquisition:** Faizel Osman.

**Investigation:** Nakul Chandan, Violet Matthews, Hejie He, Thomas Lachlan, Ven Gee Lim, Shivam Joshi, Siew Wan Hee, Angela Noufaily, Edward Parkes, Shilpa Patel, Lazaros Andronis, Joanna Shakespeare, Helen Eftekhari, Asad Ali, Gordon McGregor, Faizel Osman.

**Methodology:** Nakul Chandan, Violet Matthews, Hejie He, Thomas Lachlan, Ven Gee Lim, Siew Wan Hee, Angela Noufaily, Edward Parkes, Shilpa Patel, Lazaros Andronis, Joanna Shakespeare, Helen Eftekhari, Asad Ali, Gordon McGregor, Faizel Osman.

**Project administration:** Nakul Chandan, Violet Matthews, Thomas Lachlan, Ven Gee Lim, Shivam Joshi, Edward Parkes, Faizel Osman.

**Resources:** Nakul Chandan, Hejie He, Shivam Joshi, Helen Eftekhari, Gordon McGregor, Faizel Osman.

**Software:** Faizel Osman.

**Supervision:** Joanna Shakespeare, Asad Ali, Gordon McGregor, Faizel Osman.

**Validation:** Nakul Chandan, Hejie He, Thomas Lachlan, Faizel Osman.

**Visualization:** Nakul Chandan, Violet Matthews, Faizel Osman.

**Writing – original draft:** Nakul Chandan, Ven Gee Lim, Faizel Osman.

**Writing – review & editing:** Nakul Chandan, Violet Matthews, Hejie He, Thomas Lachlan, Ven Gee Lim, Shivam Joshi, Siew Wan Hee, Angela Noufaily, Edward Parkes, Shilpa Patel, Lazaros Andronis, Joanna Shakespeare, Helen Eftekhari, Asad Ali, Gordon McGregor, Faizel Osman.

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
