## [Decision Letter · Decision Letter 0]

30 Jul 2024

PONE-D-24-22167Cardiac prehabilitation, rehabilitation and education in first-time atrial fibrillation (AF) ablation (CREED AF): study protocol for a randomised controlled trialPLOS ONE

Dear Dr. Osman,

Thank you for submitting your manuscript to PLOS ONE. After careful consideration, we feel that it has merit but does not fully meet PLOS ONE’s publication criteria as it currently stands. Therefore, we invite you to submit a revised version of the manuscript that addresses the points raised during the review process.

**ACADEMIC EDITOR: Please insert comments here and delete this placeholder text when finished.<dear authors="">**

**Please submit your revised manuscript by Sep 13 2024 11:59PM. If you will need more time than this to complete your revisions, please reply to this message or contact the journal office at plosone@plos.org. When you're ready to submit your revision, log on to https://www.editorialmanager.com/pone/ and select the 'Submissions Needing Revision' folder to locate your manuscript file**.

**Please include the following items when submitting your revised manuscript:</dear>****A rebuttal letter that responds to each point raised by the academic editor and reviewer(s). You should upload this letter as a separate file labeled 'Response to Reviewers'.****A marked-up copy of your manuscript that highlights changes made to the original version. You should upload this as a separate file labeled 'Revised Manuscript with Track Changes'.****An unmarked version of your revised paper without tracked changes. You should upload this as a separate file labeled 'Manuscript'.******If applicable, we recommend that you deposit your laboratory protocols in protocols.io to enhance the reproducibility of your results. Protocols.io assigns your protocol its own identifier (DOI) so that it can be cited independently in the future. For instructions see: https://journals.plos.org/plosone/s/submission-guidelines#loc-laboratory-protocols. Additionally, PLOS ONE offers an option for publishing peer-reviewed Lab Protocol articles, which describe protocols hosted on protocols.io. Read more information on sharing protocols at https://plos.org/protocols?utm_medium=editorial-email&utm_source=authorletters&utm_campaign=protocols**.

**We look forward to receiving your revised manuscript.**

**Kind regards,**

**Luigi Sciarra**

**Academic Editor**

**PLOS ONE**

https://www.ncbi.nlm.nih.gov/pmc/articles/PMC6020195/?report=abstract&tool=pmcentrez

https://www.birmingham.ac.uk/Documents/college-mds/trials/bctu/A-Renovascular/Basil-3-Protocol-v4.0-06-AUG-2019.pdf

In your revision ensure you cite all your sources (including your own works), and quote or rephrase any duplicated text outside the methods section. Further consideration is dependent on these concerns being addressed.

4. In the online submission form, you indicated that [No datasets were generated or analysed during the current study. All relevant data from this study will be made available upon study completion.]. 

**5. Please review your reference list to ensure that it is complete and correct. If you have cited papers that have been retracted, please include the rationale for doing so in the manuscript text, or remove these references and replace them with relevant current references. Any changes to the reference list should be mentioned in the rebuttal letter that accompanies your revised manuscript. If you need to cite a retracted article, indicate the article’s retracted status in the References list and also include a citation and full reference for the retraction notice.**

**Additional Editor Comments:**

**Dear authors, after a careful evaluation of your manuscript and according to reviewers' suggestions our opinion is that it needs to be reviewed. We will be glad to evaluate your revised manuscript.**

**Best regards**

****

**Reviewers' comments:**

**Reviewer's Responses to Questions**

**Comments to the Author**

**1. Does the manuscript provide a valid rationale for the proposed study, with clearly identified and justified research questions?**

**The research question outlined is expected to address a valid academic problem or topic and contribute to the base of knowledge in the field.**

**Reviewer #1: Yes**

**Reviewer #2: Yes**

**2. Is the protocol technically sound and planned in a manner that will lead to a meaningful outcome and allow testing the stated hypotheses?**

**The manuscript should describe the methods in sufficient detail to prevent undisclosed flexibility in the experimental procedure or analysis pipeline, including sufficient outcome-neutral conditions (e.g. necessary controls, absence of floor or ceiling effects) to test the proposed hypotheses and a statistical power analysis where applicable. As there may be aspects of the methodology and analysis which can only be refined once the work is undertaken, authors should outline potential assumptions and explicitly describe what aspects of the proposed analyses, if any, are exploratory.**

**Reviewer #1: Yes**

**Reviewer #2: Yes**

**3. Is the methodology feasible and described in sufficient detail to allow the work to be replicable?**

**Descriptions of methods and materials in the protocol should be reported in sufficient detail for another researcher to reproduce all experiments and analyses. The protocol should describe the appropriate controls, sample size calculations, and replication needed to ensure that the data are robust and reproducible.**

**Reviewer #1: Yes**

**Reviewer #2: Yes**

**4. Have the authors described where all data underlying the findings will be made available when the study is complete?**

**The PLOS Data policy requires authors to make all data underlying the findings described in their manuscript fully available without restriction, with rare exception, at the time of publication. The data should be provided as part of the manuscript or its supporting information, or deposited to a public repository. For example, in addition to summary statistics, the data points behind means, medians and variance measures should be available. If there are restrictions on publicly sharing data—e.g. participant privacy or use of data from a third party—those must be specified.**

**Reviewer #1: Yes**

**Reviewer #2: Yes**

**5. Is the manuscript presented in an intelligible fashion and written in standard English?**

**PLOS ONE does not copyedit accepted manuscripts, so the language in submitted articles must be clear, correct, and unambiguous. Any typographical or grammatical errors should be corrected at revision, so please note any specific errors here.**

**Reviewer #1: Yes**

**Reviewer #2: Yes**

**6. Review Comments to the Author**

**Please use the space provided to explain your answers to the questions above and, if applicable, provide comments about issues authors must address before this protocol can be accepted for publication. You may also include additional comments for the author, including concerns about research or publication ethics**.

**You may also provide optional suggestions and comments to authors that they might find helpful in planning their study**.

**(Please upload your review as an attachment if it exceeds 20,000 characters)**

**Reviewer #1: PONE-D-24-22167: statistical review**

**This is a study protocol for a randomised controlled trial to assess clinical-efficacy/cost-effectiveness of prehabilitation/rehabilitation/patient-education among subjects undergoing first time atrial fibrillation ablation. The protocol relies on clearly identified research questions that can be tested by the suggested primary and secondary outcomes. The methods are standard and reproducible, sample size calculations rely on clearly stated hyptheses and methods for possible missing values are declared. Overall, I believe that this is a well-designed study protocol.**

**Reviewer #2: Authors presented an ambitious project to evaluate whether supervised sports practice, before and after an atrial fibrillation ablation, can determine differences in terms of outcome**.

**Study design seem to be appropriate for the purpose**.

**A question can arise: have authors considered that some athletes, suffering from AF, can be enrolled in the study? Can this occurrence introduce a bias in results interpretation?**

**Please specify this issue.**

**7. PLOS authors have the option to publish the peer review history of their article (what does this mean?). If published, this will include your full peer review and any attached files**.

**If you choose “no”, your identity will remain anonymous but your review may still be made public**.

**Reviewer #1: No**

**Reviewer #2: No**

****

**While revising your submission, please upload your figure files to the Preflight Analysis and Conversion Engine (PACE) digital diagnostic tool, https://pacev2.apexcovantage.com/. PACE helps ensure that figures meet PLOS requirements. To use PACE, you must first register as a user. Registration is free. Then, login and navigate to the UPLOAD tab, where you will find detailed instructions on how to use the tool. If you encounter any issues or have any questions when using PACE, please email PLOS at figures@plos.org. Please note that Supporting Information files do not need this step.**

---

## [Author Response · Author response to Decision Letter 0]

19 Aug 2024

ACADEMIC EDITOR: 

• A rebuttal letter that responds to each point raised by the academic editor and reviewer(s). You should upload this letter as a separate file labeled 'Response to Reviewers'. completed

• A marked-up copy of your manuscript that highlights changes made to the original version. You should upload this as a separate file labeled 'Revised Manuscript with Track Changes'. completed

• An unmarked version of your revised paper without tracked changes. You should upload this as a separate file labeled 'Manuscript'. completed

If applicable, we recommend that you deposit your laboratory protocols in protocols.io to enhance the reproducibility of your results. Protocols.io assigns your protocol its own identifier (DOI) so that it can be cited independently in the future. We can perform this.

For instructions see: https://journals.plos.org/plosone/s/submission-guidelines#loc-laboratory-protocols. Additionally, PLOS ONE offers an option for publishing peer-reviewed Lab Protocol articles, which describe protocols hosted on protocols.io. Read more information on sharing protocols at https://plos.org/protocols?utm_medium=editorial-email&utm_source=authorletters&utm_campaign=protocols.

We look forward to receiving your revised manuscript.

Kind regards,

Luigi Sciarra

Academic Editor

PLOS ONE

1. Please ensure that your manuscript meets PLOS ONE's style requirements, including those for file naming. The PLOS ONE style templates can be found at Completed

Completed

https://www.ncbi.nlm.nih.gov/pmc/articles/PMC6020195/?report=abstract&tool=pmcentrez

We have now cited this paper in our manuscript (page 5)

https://www.birmingham.ac.uk/Documents/college-mds/trials/bctu/A-Renovascular/Basil-3-Protocol-v4.0-06-AUG-2019.pdf

We have now referenced this protocol in our manuscript (page 15)

In your revision ensure you cite all your sources (including your own works), and quote or rephrase any duplicated text outside the methods section. Further consideration is dependent on these concerns being addressed.

3. We note that the grant information you provided in the ‘Funding Information’ and ‘Financial Disclosure’ sections do not match. We have declared funding received from Boston Scientific for this study in our manuscript (page 18) and in the Funding Information section during online submission. 

When you resubmit, please ensure that you provide the correct grant numbers for the awards you received for your study in the ‘Funding Information’ section. No grant number has been provided to us by the funder

4. In the online submission form, you indicated that [No datasets were generated or analysed during the current study. All relevant data from this study will be made available upon study completion.]. 

This policy applies to all data except where public deposition would breach compliance with the protocol approved by your research ethics board. If your data cannot be made publicly available for ethical or legal reasons (e.g., public availability would compromise patient privacy), please explain your reasons on resubmission and your exemption request will be escalated for approval. We confirm all data will be freely available 

5. Please review your reference list to ensure that it is complete and correct. If you have cited papers that have been retracted, please include the rationale for doing so in the manuscript text, or remove these references and replace them with relevant current references. Any changes to the reference list should be mentioned in the rebuttal letter that accompanies your revised manuscript. If you need to cite a retracted article, indicate the article’s retracted status in the References list and also include a citation and full reference for the retraction notice. This has now been checked and completed

Reviewer #1: PONE-D-24-22167: statistical review

This is a study protocol for a randomised controlled trial to assess clinical-efficacy/cost-effectiveness of prehabilitation/rehabilitation/patient-education among subjects undergoing first time atrial fibrillation ablation. The protocol relies on clearly identified research questions that can be tested by the suggested primary and secondary outcomes. The methods are standard and reproducible, sample size calculations rely on clearly stated hyptheses and methods for possible missing values are declared. Overall, I believe that this is a well-designed study protocol. Thank you

Reviewer #2: Authors presented an ambitious project to evaluate whether supervised sports practice, before and after an atrial fibrillation ablation, can determine differences in terms of outcome.

Study design seem to be appropriate for the purpose. Thank you

A question can arise: have authors considered that some athletes, suffering from AF, can be enrolled in the study? Can this occurrence introduce a bias in results interpretation?

Please specify this issue.

The question about athletes, particularly high-endurance level athletes, did come to mind when planning and developing the research protocol for CREED AF. The general consensus was to maintain generalizability of the study by being inclusive to as many individuals suffering with atrial fibrillation (AF) undergoing first time AF ablation as possible. 

Concern may rest with higher than average VO2 peak seen in these individuals when matched with non-athletes. It has been observed that high-endurance athletes do also suffer with AF, with reduced physical ability/fitness prior to ablation, with improvement of most to near previous levels of fitness/exertion after ablation (Liu et al., 2022). The application of the CREED AF intervention, which aims at delivering a moderate level of exercise (tailored to the individual) alongside risk factor management, is equally applicable in these potential participants. Furthermore, the primary outcome measure is mean change in VO2 peak from baseline to primary follow-up at 10 +/-2 weeks post-ablation in the two groups; measuring and comparing the change would minimize any bias, if any, derived by inclusion of high-endurance athletes.

We have now included this in the discussion section on page 17

---

## [Decision Letter · Decision Letter 1]

10 Sep 2024

Cardiac prehabilitation, rehabilitation and education in first-time atrial fibrillation (AF) ablation (CREED AF): study protocol for a randomised controlled trial

PONE-D-24-22167R1

Dear Dr. Osman,

We’re pleased to inform you that your manuscript has been judged scientifically suitable for publication and will be formally accepted for publication once it meets all outstanding technical requirements.

Kind regards,

Luigi Sciarra

Academic Editor

PLOS ONE

Additional Editor Comments (optional):

Dear authors I 'm glad to inform you that your paper has been positively evaluated.

Reviewers' comments:

Reviewer's Responses to Questions

**Comments to the Author**

1. Does the manuscript provide a valid rationale for the proposed study, with clearly identified and justified research questions?

Reviewer #1: Yes

Reviewer #2: Partly

2. Is the protocol technically sound and planned in a manner that will lead to a meaningful outcome and allow testing the stated hypotheses?

Reviewer #1: Yes

Reviewer #2: Yes

3. Is the methodology feasible and described in sufficient detail to allow the work to be replicable?

Reviewer #1: Yes

Reviewer #2: Yes

4. Have the authors described where all data underlying the findings will be made available when the study is complete?

Reviewer #1: Yes

Reviewer #2: Yes

5. Is the manuscript presented in an intelligible fashion and written in standard English?

Reviewer #1: Yes

Reviewer #2: Yes

6. Review Comments to the Author

You may also provide optional suggestions and comments to authors that they might find helpful in planning their study.

Reviewer #1: I already recommended to accept the study protocol during th efirst round of review.

Reviewer #2: Authors satisfied requests advanced.

I have no additional comments.

The protocol can be now accepted

7. PLOS authors have the option to publish the peer review history of their article (what does this mean?). If published, this will include your full peer review and any attached files.

Reviewer #1: No

Reviewer #2: **Yes: **Antonio Scarà MD

---

## [Editor Report · Acceptance letter]

24 Sep 2024

PONE-D-24-22167R1 

PLOS ONE

Dear Dr. Osman, 

I'm pleased to inform you that your manuscript has been deemed suitable for publication in PLOS ONE. Congratulations! Your manuscript is now being handed over to our production team.

Kind regards, 

on behalf of

Dr. Luigi Sciarra 

Academic Editor

PLOS ONE